# Clutching at Guidance Cues: The Integrin–FAK Axis Steers Axon Outgrowth

**DOI:** 10.3390/biology12070954

**Published:** 2023-07-03

**Authors:** Mathew Davis-Lunn, Benjamin T. Goult, Melissa R. Andrews

**Affiliations:** 1Faculty of Environmental and Life Sciences, University of Southampton, Southampton SO17 1BJ, UK; m.j.davis-lunn@soton.ac.uk; 2School of Biosciences, University of Kent, Canterbury CT2 7NJ, UK; b.t.goult@kent.ac.uk; 3Centre for Human Development, Stem Cells and Regeneration, School of Biological Sciences, University of Southampton, Southampton SO17 1BJ, UK

**Keywords:** integrin, focal adhesion kinase, neurite outgrowth, axon regeneration, adhesion complex, point contact, growth cone, talin, kindlin, extracellular matrix

## Abstract

**Simple Summary:**

Neurons are biological wires that send signals not only across the brain, but throughout the body. Like almost all cells in the human body, neurons adhere to their surroundings through receptors known as integrins. As the nervous system develops, integrins become essential for neuronal wires, known as axons, to adhere within their environment and pull themselves forward as they grow towards their target. Following injury to the nervous system, the damaged axons must regrow towards their target to restore communication. The integrin receptor therefore plays an important role in this process of regeneration. Integrins mediate their effects through an intricate scaffold of proteins inside the cell, which must be coordinated by a specific central protein, called focal adhesion kinase. This review discusses the signalling pathway of integrin receptors and focal adhesion kinase, specifically in neurons, and how this signalling pathway mediates neuronal growth. We also discuss how integrins and focal adhesion kinase could contribute to neuronal repair following nervous system injury.

**Abstract:**

Integrin receptors are essential contributors to neurite outgrowth and axon elongation. Activated integrins engage components of the extracellular matrix, enabling the growth cone to form point contacts, which connect the extracellular substrate to dynamic intracellular protein complexes. These adhesion complexes facilitate efficient growth cone migration and neurite extension. Major signalling pathways mediated by the adhesion complex are instigated by focal adhesion kinase (FAK), whilst axonal guidance molecules present in vivo promote growth cone turning or retraction by local modulation of FAK activity. Activation of FAK is marked by phosphorylation following integrin engagement, and this activity is tightly regulated during neurite outgrowth. FAK inhibition slows neurite outgrowth by reducing point contact turnover; however, mutant FAK constructs with enhanced activity stimulate aberrant outgrowth. Importantly, FAK is a major structural component of maturing adhesion sites, which provide the platform for actin polymerisation to drive leading edge advance. In this review, we discuss the coordinated signalling of integrin receptors and FAK, as well as their role in regulating neurite outgrowth and axon elongation. We also discuss the importance of the integrin–FAK axis in vivo, as integrin expression and activation are key determinants of successful axon regeneration following injury.

## 1. Introduction

Neurons are recognisable by their characteristically polarised structure, projecting dendrites and an axon from the cell soma. The axon is a specialised projection enabling the transduction of electrochemical signals over long distances up to hundreds of millimetres, communicating with dendrites of connected downstream neurons. During the development of the nervous system, axon growth is driven and directed from the distal tip by use of a ‘growth cone’ rich in actin [1]. A neuronal growth cone (shown in Figure 1) is divisible into central and peripheral domains, the latter harbouring the lamellipodia, which is a network of branched actin filaments at the leading edge. Regulated bundling of these actin filaments promotes outward protrusion of the membrane to form micrometre-length filopodia [2]. These filopodial ‘fingers’ extending from the lamellipodial ‘hand’ then seek guidance cues in the extracellular matrix (ECM). The growth cone utilises receptors known as integrins to adhere to the ECM and pull the growth cone and attached axon forward, effectively extending the length of the axon. Integrin adhesion sites in neuronal cells take on specialised roles and are known as point contacts [3].

When not actively engaged with the ECM, the actin cytoskeleton undergoes treadmilling. Treadmilling is a process resulting from the continual polymerisation of actin filaments at membrane-proximal barbed ends, which encounters resistance or ‘push-back’ from the membrane and results in net retrograde flow of the actin network. In the growth cone of developing or regenerating axons, point contacts support axonal outgrowth by providing an anchor point or ‘clutch’, which can slow the retrograde flow of actin [4,5,6]. This anchoring allows the polymerisation of actin filaments at the growth cone lamellipodia and filopodial tip to exert sufficient force upon the membrane to deform it and expand the reach of the cell [7]. Both actin polymerisation [8] and the formation of filopodia [9] are essential for efficient neurite outgrowth. Point contact formation is dependent upon the presence of positive ECM guidance cues, which can therefore influence the direction of growth through the integrin–substrate interaction, enabling local actin clutching. In addition, negative guidance cues can promote growth cone turning or axonal retraction by locally destabilising point contacts [10].

The direct interaction between the ECM and intracellular point contact machinery is dependent upon the presence of transmembrane integrin receptors. Integrins engage the ECM via positive guidance cues and facilitate the construction of large and dynamic intracellular adhesion complexes (Figure 2D). Integrin adhesion complexes (IACs) come in many forms and coordinate many cellular functions. IAC composition is dependent upon the integrin subunit present, properties of the ECM such as rigidity and ligand density, and cell type. Over 900 proteins have been identified at IACs [11]. Neuronal point contacts are undoubtably specialised structures; however, many key proteins are conserved between IACs. For example, non-neuronal focal adhesions have been recently reviewed in greater detail [12] with further reviews focused on proteins relevant in neuronal point contacts such as paxillin [13], kindlin [14], talin [15] and vinculin [16]. An essential signal transducer, focal adhesion kinase (FAK) forms a key structural hub of the adhesion complex and coordinates adhesion turnover [17]. Furthermore, phosphorylation of FAK is commonly used as a readout for integrin activation and engagement with an extracellular substrate [18,19,20]. In this review, we discuss the role of the integrin–FAK axis in neurite outgrowth and its implication for axon repair. To our knowledge, this review is the first exploring the function of FAK in neurite outgrowth and its synergistic role with the integrin receptor, with reference to the mechanism by which IACs support neurite outgrowth. Readers are referred to the following reviews for further structural detail on integrins [21,22] and FAK [23].

## 2. Integrin Structure and Activation

Integrin receptors are heterodimeric, formed from a pool of 26 mammalian genes comprising 18 α- and 8 β-subunits. Combinations of the α-subunit in dimerization with specific β-subunits total 24 integrin receptors and dictate the receptors’ preference for a variety of ECM glycoproteins which act as integrin ligands and guidance cues, for example laminin or fibronectin [24,25]. Each integrin heterodimer exhibits similar architecture (Figure 2A), divisible into intracellular, transmembrane, and extracellular regions. In the typical extracellular structure from the membrane outwards, the α-integrin subunit contains two calf domains, a connecting thigh domain, and a β-propeller [26]. The β-integrin subunit is typically formed of a β-tail domain, four EGF-like repeats, an Ig-like domain and an I/A domain containing a metal ion-dependent adhesion site (MIDAS) [26]. In the heterodimeric structure, the β-propeller and I/A domains complex to form the N-terminal integrin ‘head’, which contains the ligand binding site [26,27]. The α- and β-subunits extend two parallel legs from the connected head, into the membrane. The precise structure and activation mechanism of integrin heterodimers is dependent upon the incorporation of specific subunits; as such, the following discussion provides a general model rather than a definitive guide.

The plasma membrane restrains the integrin receptor into a bent, inactive conformation (Figure 2B), which is stabilised by intramolecular interactions between the EGF2 and 3 domains in the β-subunit [26,28], in addition to a salt bridge adjoining the two cytoplasmic tails [29]. The classical mechanism of integrin activation requires the binding of two endogenous proteins, talin and kindlin, to the cytoplasmic tail of the β-subunit, which may occur either sequentially or simultaneously [25]. Talin contains a FERM (4.1-ezrin-radixin-moesin homology) domain head coupled to a rod domain tail, formed of successive alpha helix bundles [30]. The talin FERM domain binds a membrane-proximal NPxY motif [31], breaking the salt bridge between the α- and β-integrin tails [32]. Expression of the talin head alone in adult rat dorsal root ganglion (DRG) neurons is sufficient to activate integrin (in the presence of endogenous kindlin); however, the tail domain is required to promote neurite outgrowth as a downstream output [33]. Neurite outgrowth is likely to require key interactions between the talin rod and the adhesion complex and cytoskeleton [34]. Kindlin collaboratively binds a second, membrane-distal, NPxY motif through its own FERM domain; however, in the absence of talin, kindlin alone may be insufficient for integrin activation [35]. The synergistic binding of both kindlin and talin weakens the interactions restraining the integrin receptor and allows the extracellular integrin domains to extend (Figure 2C), increasing affinity for their preferred ECM substrate. When co-expressed in HT1080 cells, kindlin and talin induce greater cell-wide activation of integrins than expression of either molecule alone [35].

## 3. Outside-In Integrin Engagement

Activation of the integrin receptor by kindlin and talin (Figure 2C) is referred to as ‘inside-out’ signalling [36]. ‘Outside-in’ signalling (Figure 2D), is induced most efficiently when signals from the intracellular and extracellular environments act in synergy, though binding of the ECM ligand may precede inside-out activation [37]. The activated integrin receptor attaches to the ECM through binding of the integrin head to specific recognition sites on the glycoprotein, for example laminin [38]. Most commonly, integrins recognise the presence of an RGD (arginine-glycine-aspartic acid) motif found in many ECM glycoproteins; however, additional motifs have been identified in specific integrin ligands such as the LDV (leucine-aspartic acid-valine) motif of VCAM-1 [39]. Ligand binding is also dependent on a number of additional metal ion interactions, exemplifying the multiplicity of signals required to collaborate for efficient integrin signalling [40]. Our understanding of downstream adhesion signalling is largely derived from studies on mature focal adhesions in vitro, where multivalent binding to extracellular ligands induces integrin clustering to facilitate intracellular scaffolding of large focal adhesion complexes. In comparison to the well-characterised adhesions that form on two-dimensional substrates coated on glass, the contribution of receptor clustering in the formation of smaller neuronal point contacts has not been as well studied, nor has the contribution of the three-dimensional and complex ECM in vivo. However, from an intracellular perspective, receptor clustering could be aided by kindlin [35] and other adhesion proteins known to form multimers at the membrane such as talin and FAK.

The adhesion complex is constructed from an intricate network of protein–protein and protein–lipid interactions, and is assisted by membrane-targeting complexes, exemplified by the Rap1-RIAM interaction which brings talin to the membrane [41]. The network of interactions responsible for FAK recruitment is typically complex, dependent on membrane phospholipids such as PI(4,5)P2, and multiple proteins including kindlin and p190RhoGEF [42]. FAK can in fact be a precursor to talin [43] and/or paxillin [44] recruitment; hence, it is likely that PI(4,5)P2 instigates recruitment of FAK to the membrane prior to further scaffolding interactions [26]. However, the stepwise mechanism by which FAK is bound to the adhesion complex remains unclear.

## 4. FAK Signalling

FAK is a kinase consisting of three major domains: the N-terminal FERM domain, the C-terminal focal adhesion targeting (FAT) domain, and the central kinase domain (Figure 3). The FERM domain is formed of three lobes of approximately 100 amino acids each, that give a characteristic cloverleaf structure [45]. The FAT domain is a four-helix bundle, containing sites for key interactions with other adhesion proteins including paxillin [46] and talin [43]. The central kinase domain of FAK contains two lobes forming a typical kinase fold, in addition to a regulatory loop. Tyrosine residue 397 at the periphery of this kinase domain is a key signalling residue, phosphorylation of which is used as the canonical identifier of FAK activation.

In neuronal isoforms of FAK, an additional PWR (proline-tryptophan-arginine) insert of unknown function is present within the FAT domain, denoted as FAK+ [47,48]. Further combinations of inserts may be present either side of Y397, which contribute to FAK activation. These inserts are termed box 6 and box 7, where the presence of both is denoted as FAK+6,7. The distribution of the resulting neuronal FAK isoforms, FAK+, FAK+6, FAK+7 and FAK+6,7, varies throughout brain tissue [49].

A second member of the adhesion kinase family is protein tyrosine kinase 2β (Pyk2), with a similar domain structure to FAK. Despite the homology between Pyk2 and FAK, Pyk2 has largely been studied in the regulation of dendritic spines, with relatively little characterization at IACs in developing or regenerating axons [50]. In rat E18 hippocampal neurons plated on poly-L-lysine, Pyk2 localizes to the central domain of the growth cone, but not the peripheral domain or lamellipodia where FAK is present [51]. Pyk2 expression increases following the development of the central nervous system (CNS), and Pyk2 knockout mice develop normal CNS anatomy [52]. These data suggest a non-essential role for Pyk2 in axon outgrowth, hence this review has focused upon the integrin–FAK signalling axis.

Canonical FAK is autoinhibited by an intramolecular interaction between the F2 lobe of the FERM domain and the C lobe of the kinase domain (Figure 4A) [53]. The release of this autoinhibitory interaction is supported by the binding of basic regions of the FAK FERM domain to membrane PI(4,5)P2, alongside the formation of FAK dimers [23]. Initial FAK dimerization may be mediated either by tryptophan residue 266 (W266) within the FERM domain, or a second interface between the FERM and FAT domains of opposing molecules which is stabilised by paxillin [54]. FAK is also able to form oligomers at the membrane through the W266 interaction, which likely supports more efficient FAK activation in nascent adhesion sites [23,54]. Additionally, in mature adhesion sites, talin can bind the FAT domain of FAK and recruit further FAK molecules; however, this is less relevant in forming nascent adhesions, and hence likely also point contacts [43]. FAK dimerization results in release of the FERM–kinase domain interaction and enables trans-phosphorylation on Y397, activating FAK (Figure 4B). In neuronal isoforms, however, the presence of boxes 6 and 7 results in a higher basal level of FAK pY397 [49]. This higher basal activity derives from each insert, contributing to a reduction in autoinhibition by the FERM domain, and isoforms containing box 7 can instead autophosphorylate in *cis* [55]. Whilst research has begun to explore the importance of these inserts for synaptic activities of FAK, their contribution to axonal outgrowth is not known. It is possible that the increased basal activity of neuronal FAK isoforms plays a role in the migrating growth cone, allowing rapid recruitment into nascent adhesion sites and faster turnover of point contacts.

Once phosphorylated, FAK Y397 recruits Src via the Src SH2 domain. Src recruitment permits phosphorylation of FAK Y576 and Y577 which reside in the kinase regulatory loop, reducing FERM domain autoinhibition in canonical FAK isoforms and further increasing FAK autophosphorylation upon Y397 [53,56]. The constitutively active ‘superFAK’ mutant (FAK K578E and K581E) likely induces a similar effect to increase FAK autophosphorylation, though downstream signalling remains dependent upon cell adhesion [57]. The relevance of the regulatory loop in neuronal FAK isoforms, which are already less prone to autoinhibition, has not been studied.

Src also phosphorylates FAK on Y861 and Y925 to create binding sites for further scaffolding proteins (Figure 4C). pY861 recruits p130Cas, facilitated by proline-rich regions of FAK binding the p130Cas SH3 domains [58,59]. Following its recruitment, phosphorylation of p130Cas by Src recruits the Crk/DOCK180/Rac1 complex which promotes local membrane protrusion [10,60]. FAK pY925 accelerates turnover of adhesion complexes by recruitment of Grb2, which may competitively displace FAK from paxillin [61]. Grb2 also induces the Ras/ERK2 pathway to phosphorylate myosin light-chain kinase, promoting contractility [62], and this pathway could be a minor contributor to adhesion turnover as contractile force can accelerate the disintegration of poorly adhered contacts [7]. Interestingly, the turnover rate of adhesions in *Xenopus* spinal neurons can be decreased by expression of a non-phosphorylatable FAK Y925F mutant [63]; conversely, the expression of the phosphomimetic FAK Y925E enhances the rate of membrane protrusion in mouse embryonic fibroblasts [64].

Besides this core signalling pathway, further direct and indirect interactions with FAK are likely to contribute to cellular motility. For example, FAK interacts with the Arp2/3 complex to promote lamellipodial remodelling and leading edge advance [65]. Additionally, a direct interaction between FAK and dynamin2 is implicated in clathrin-dependent endocytosis of focal adhesions during turnover [66]. Endocytosed IACs have been shown to retain integrins in an active conformation. Disassembly of focal adhesions can be forced using nocodazole treatment, resulting in their endocytosis. When nocodazole treatment is combined with pharmacological inhibition of FAK, a decreased proportion of active β1 integrin is present in early endosomes [67]. This study suggests FAK may help impart conformational memory onto endocytosed integrins and could directly enable more efficient cellular migration by retaining integrins in activated states, ready for presentation once returned to the cell surface.

In summary, activated FAK coordinates multiple signalling outputs to induce the turnover of point contacts, and could retain neuronal integrins in an active state for rapid recycling. These functions indicate that FAK is an important component of efficient growth cone advance during neurite outgrowth.

## 5. Regulation of Neurite Outgrowth by FAK

Several studies have targeted FAK in the context of neurite outgrowth. Principally, FAK knockdown using miRNA inhibits neurite outgrowth in differentiating PC12 cells when plated on laminin, fibronectin or collagen [68]. Likewise, shRNA knockdown or pharmacological inhibition of FAK also inhibited outgrowth of postnatal mouse hippocampal neurons when plated on poly-L-lysine [69]. Similarly, expression of the dominant negative isoform of FAK, FAK-related non-kinase (FRNK), can suppress neurite outgrowth of adult rat DRG neurons treated with soluble laminin and NGF [70]. Studies in Xenopus neural tube explants show that both FAK knockdown and FRNK overexpression impede neurite outgrowth by slowing point contact turnover [63,71].

Conditional deletion of FAK from mouse hippocampal neurons from in vitro day 3 (DIV3) likewise results in a slower rate of neurite outgrowth [72]. Interestingly, between deletion at DIV3 and analysis at DIV6, this study observed an increase in the total neurite length per cell, despite a slower growth rate of individual neurites. Studies since have suggested this aberrant overgrowth of hippocampal branches after FAK deletion resulted from an impairment of branch pruning following the early phase of neurite outgrowth [72]. Therefore, FAK plays a biphasic role in arborisation of hippocampal neurons, first enabling the outgrowth of neurites before promoting branch pruning. FAK uses mechanisms of point contact turnover described above to promote neurite retraction, by disassembling adhesion sites and inducing growth cone collapse [73,74]. Additionally, in mouse hippocampal neurons, FAK activation is also essential for outgrowth of dendrites [75] and dendritic spines [69]. Contrastingly however, FAK activation is also required for dendritic spine maintenance [76], suggesting that the effects of FAK activation are dependent on subcellular location.

FAK can provide a context-specific accelerator or brake to growth cone advance, but also applies these functions locally to steer axonal guidance. When the growth cone encounters a positive guidance cue, adhered point contacts clutch the retrograde flow of actin, and downstream signalling promotes local protrusion [5]. Soluble guidance cues, however, require a different mechanism of guidance as their binding cannot provide traction. Brain-derived neurotrophic factor (BDNF) locally stimulates FAK activity to induce point contact turnover and growth cone turning, whilst neurons expressing FAK Y925F show a deficient response to BDNF in part due to a reduced adhesion turnover rate [63]. Some repulsive guidance cues also utilise FAK; for example, Sema3A repels the growth cone by stimulating FAK Y925 phosphorylation [74]. Comparing these examples of BDNF and Sema3A, it is unclear how both a positive and negative soluble guidance cue can utilise the same mechanism of FAK-induced adhesion turnover to direct the growth cone in opposite directions. It is likely that the complexity of FAK signalling hides key, possibly temporal, differences in the response to each of these cues.

## 6. Targeting the Integrin–FAK Axis

The importance of FAK for efficient neurite outgrowth is evident, but can FAK therefore act as a target to enhance outgrowth? Overexpression of FAK in postnatal mouse hippocampal neurons lengthened the total neurite length per cell from DIV0 to DIV1; however, whilst this was reflected by an increase in neurite branching, the length of individual neurites was not measured [69]. Similarly, overexpressing signalling partner Src in vivo at E15.5 resulted in an aberrant branching of cortical plate neurons via FAK pY925 signalling [77]. These results are both taken in the context of early neurite outgrowth, whereas in later stages of neuronal development, FAK appears to switch to a pruning function, when unwanted branches are no longer sufficiently stimulated [72]. Expression of FAK constructs intended to overstimulate (superFAK) or inhibit (FAK Y861F, Y925F, FRNK) FAK activity all appear to disrupt the rate of neurite outgrowth in *Xenopus* explant cultures; however, the total neurite length and branching effect was not shown or discussed in these papers [63,78].

In most studies to date, cDNA for exogenous expression of FAK has utilised the canonical FAK isoform rather than the neuronal isoforms. Based on the function of its additional inserts, we speculate that neuronal FAK is optimised for recruitment to, and activation at, point contacts. For example, isoform specific inserts may induce key temporal differences in downstream signalling, or induce structural changes to the adhesion scaffold. To our knowledge, no published data are available comparing the cellular function of neuronal FAK isoforms, and their contribution to neurite outgrowth remains to be studied.

The role of FAK in guiding neurite outgrowth is dependent upon the expression of an appropriate integrin receptor to engage the ECM. Both integrin expression and ECM composition are dependent on the developmental stage of the nervous system. For example, adult rat DRG neurons exhibit poor growth on low concentrations of both laminin and fibronectin in comparison to DRGs isolated from P0 pups, a phenotype rescued by exogenous expression of integrin subunits binding laminin (α1), or fibronectin (α5), respectively [79]. Further in vitro evidence similarly demonstrates the requirement for appropriate integrin expression in both adult DRG neurons [80] and hippocampal neurons [81] responding to specific ECM stimuli. References to studies targeting specific integrin heterodimers in this review are summarised in Table 1. Readers are referred to comprehensive overviews of integrin heterodimer substrate and function for further detail [24,82].

Exogenous expression of integrin subunits therefore represents a therapeutic opportunity to promote axonal regeneration in vivo. In the adult spinal cord, CNS expression of permissive substrates such as laminin becomes limited relative to development [85,86]. Following spinal cord injury (SCI), the CNS exhibits a more inhibitory ECM and an upregulation of chondroitin sulphate proteoglycans (CSPGs). One glycoprotein upregulated following injury, tenascin-C, is recognised by the α9β1 integrin heterodimer [87]. α9 integrin is not expressed in adult neurons; however, exogenous expression of α9 integrin has been used to enhance the regenerative response in vivo following a dorsal root and dorsal column crush injury in adult rats [83]. However, integrin subunits are excluded from some mature CNS axons, minimising their contribution to axonal growth [88]. The targeting of molecules that are responsible for delivery of integrin subunits to the axonal growth cone is emerging as a strategy to mitigate this deficit [67,89,90]. Localisation of specific integrin receptors and their contribution to axonal regeneration has been recently reviewed in great detail [82], and readers are referred to an additional review on targeting integrin trafficking in the context of axon regeneration [91]. The use of integrin gene therapy for SCI is therefore a promising treatment strategy, though it is currently limited by our ability to deliver this critical receptor to the axonal growth cone.

Aggrecan, a CSPG expressed by mature neurons and upregulated following SCI, has been shown in vitro to inhibit neurite outgrowth of adult rat DRGs in the presence of laminin [18]. CSPG-mediated inhibition has also been demonstrated in both adult rat DRGs [92] and postnatal rat retinal ganglion cells [93] when plated on the CSPGs neurocan and phosphacan. Likewise, versican is repulsive to neurite outgrowth in embryonic chick DRG and retinal explants [94]. CSPGs such as aggrecan inactivate integrins, thereby reducing FAK phosphorylation in the process. Overexpression of kindlin-1 can reactivate these integrins, restore FAK activity, and improve neurite outgrowth [19,20]. Consequently, co-expression of kindlin-1 with α9 integrin following dorsal root crush injury in adult rats induces a significantly greater regeneration of axons into the CNS when compared with expression of either α9 integrin or kindlin-1 alone [20]. These studies not only demonstrate the necessity for activated integrin during axon regeneration, but also the synergy with FAK as a signalling partner. In another study, expression of the talin FERM domain alone in adult DRGs in vitro was shown to activate integrins, but simultaneously reduced the activation of FAK [33]. The talin head alone provides a dominant-negative effect, ameliorating the positive contribution of full-length talin to neurite outgrowth, and demonstrating the need for collaborative activity among FAK and other IAC proteins.

Following SCI, aggrecan inhibits integrin signalling [20]; however, the molecular mechanism of receptor inactivation is currently unclear. CSPGs are known to signal through the receptor-type tyrosine-protein phosphatase S (PTPσ) to inhibit neurite outgrowth [95]. PTPσ can localise to some adhesion sites, including focal adhesions [96], and accumulates within the immobilised growth cone [97]. The inactivation of integrins by CSPGs may therefore arise as a by-product of PTPσ engagement, either through steric hinderance, or sticking of the growth cone in a region absent of positive stimuli. Enzymatic digestion of the CSPG glycosaminoglycan side chains by chondroitinase ABC removes the ligands mediating the PTPσ interaction, improving neurite outgrowth [95] and increasing FAK activation, which is indicative of activated integrins [98].

Integrins can also be inactivated by inhibitory myelin-derived molecules, for example Nogo-A, which acts in combination with myelin associated glycoprotein (MAG) and oligodendrocyte myelin glycoprotein (OMgp), inhibiting axonal outgrowth via the Nogo receptor [99]. Specific to Nogo-A is an additional domain which can indirectly reduce activation of specific integrins (α5β1, αVβ3), though the mechanism behind this effect remains unknown [84]. MAG, however, has also been shown to directly engage β1 integrin to act as a repulsive guidance cue through local activation of FAK [100].

In summary, reactivating integrins that have been developmentally downregulated, or silenced by an inhibitory environment, provides a therapeutic opportunity to promote axonal regeneration in vivo, such as in the context of SCI. This requires that positive stimuli remain accessible in the local environment; hence, combinational therapies using both overexpression of key integrin receptors together with an integrin activator could target specific ECM stimuli upregulated following SCI. Restoring integrin-mediated adhesion can then reconnect the intracellular machinery of regenerating axons to the ECM and steer the regenerative response.

## 7. Conclusions

We have integrated the studies discussed in this review to propose a current model of the integrin–FAK signalling axis in the neuronal growth cone (Figure 5). Expression and activation of the appropriate integrin receptors are essential for adherence of the growth cone to the ECM. ECM engagement is a prerequisite for the activation of FAK, which then enables efficient turnover of point contacts by a tight regulatory system integrating mechanical and biochemical signals. FAK ensures point contacts are disassembled with the correct spatial and temporal regulation, steering the growth cone through responses to extracellular guidance cues. FAK may also retain activated integrins in internalised endosomes akin to its role in focal adhesions, contributing to efficient growth cone migration. Disrupting the balance of FAK signalling through a range of methods has been shown to perturb the efficiency of neurite outgrowth, but despite this, FAK remains an attractive centrepiece of the point contact machinery for modulating neurite outgrowth in an experimental context. Exploring the remaining gaps in our knowledge of the FAK signalling cascade may unmask a key focal point which can be exploited to further enhance neurite outgrowth. For example, by altering the temporal structure of FAK signalling to optimise cytoskeletal force generation, FAK modulation could enable the co-delivery of an additional growth-promoting mechanism alongside a specific integrin subunit. A deeper understanding of the integrin–FAK axis could therefore enhance current therapeutic approaches towards axon regeneration.

## Figures and Tables

**Figure 1 biology-12-00954-f001:**
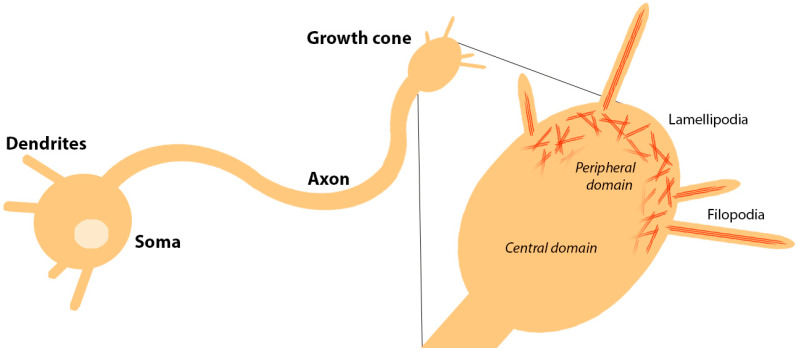
The growth cone is formed at the tip of developing axons to guide their extension. Growth cones are divisible into a central and peripheral domain, where the peripheral domain contains structures rich in actin that help to regulate motility. The lamellipodia is a network of branched actin filaments (red) which is remodelled during growth cone advance. Regulated bundling of actin filaments drives the protrusion of filopodia, which sense the environment ahead of the growth cone.

**Figure 2 biology-12-00954-f002:**
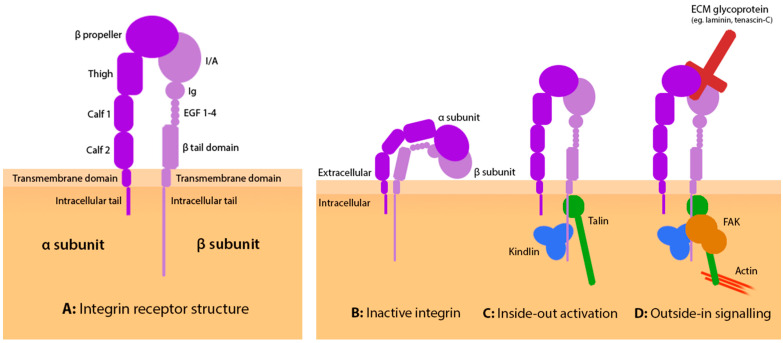
(**A**) The integrin heterodimer is formed of α- and β-subunits. The typical α-subunit (purple) is formed of a short intracellular tail, transmembrane domain, two calf domains, a thigh domain and β-propeller. The β-subunit (lilac) is commonly formed of a long intracellular tail with kindlin and talin binding sites, transmembrane domain, β-tail domain, 4 EGF-like repeats, an Ig-like domain and I/A domain. (**B**) The integrin receptor is maintained in an inactive conformation, with low affinity for ECM ligands, by intramolecular interactions between the α- and β-subunits. (**C**) Binding of the integrin activators kindlin and talin to the β-integrin tail induces conformational activation of the integrin receptor, allowing the extracellular domains to extend and reveal a binding site for glycoproteins within the ECM. This is also known as ‘inside-out’ signalling. (**D**) In ‘outside-in’ signalling, integrin binding to the favoured glycoprotein stabilises the adhesion complex, enabling the scaffolding of further adhesion proteins including focal adhesion kinase (FAK). The resulting intracellular signalling cascade coordinates adhesion turnover and growth cone migration through signalling pathways discussed in Section 4. The adhesion complex adjoins the extracellular substrate to the actin cytoskeleton, which facilitates mechanical signalling and growth cone migration.

**Figure 3 biology-12-00954-f003:**
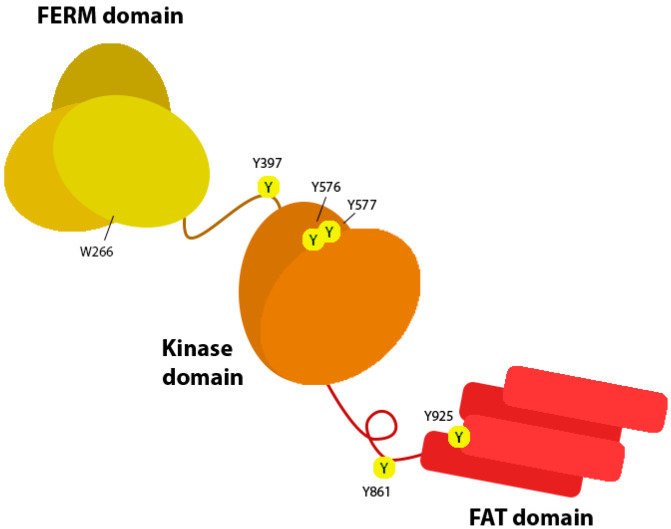
Domain structure of focal adhesion kinase (FAK). The N-terminal FERM domain (yellow) contains tryptophan residue 266 (W266) and is connected to the central kinase domain (orange) by a linker region, adjacent to which is tyrosine residue 397. Tyrosines 576 and 577 are found within the kinase domain. The C-terminal focal adhesion targeting (FAT) domain (red) contains tyrosines 861 and 925.

**Figure 4 biology-12-00954-f004:**
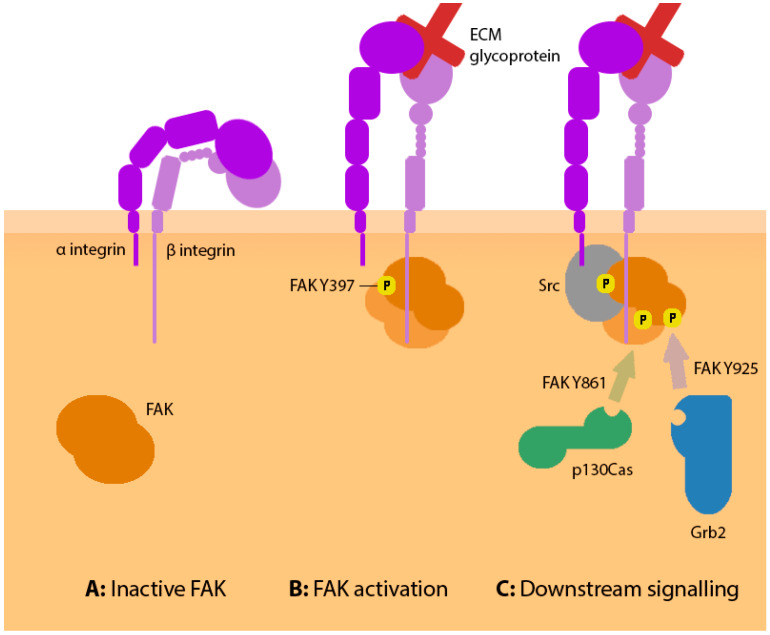
(**A**) FAK is found in the cytoplasm where shielding of the kinase domain by the FERM domain confers a state of autoinhibition. (**B**) Integrin engagement supports the recruitment of FAK to the adhesion complex. The FAK FERM domain binds the β-integrin tail, whilst further protein interactions help relieve autoinhibition. Adhesion-targeted FAK dimerises and autophosphorylates upon tyrosine residue 397. (**C**) FAK pY397 facilitates binding of Src (grey), which phosphorylates further tyrosine residues of FAK including Y861 and Y925. These residues recruit p130Cas (green) and Grb2 (blue) respectively, each inducing complementary downstream signalling pathways which contribute to the characteristic outputs of FAK activity.

**Figure 5 biology-12-00954-f005:**
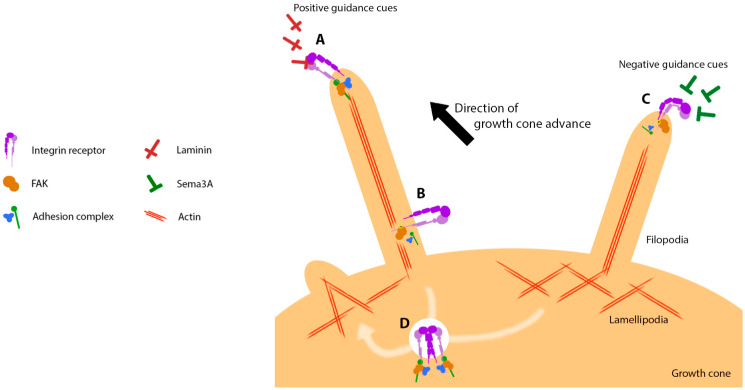
The integrin–FAK axis mediates neurite outgrowth by enabling the formation and recycling of point contacts within the peripheral domain of the growth cone. (**A**) Integrin receptors mediate growth cone adhesion to the ECM via positive guidance cues such as laminin. FAK scaffolds with the IAC underneath the engaged integrin, clutching the retrograde flow of actin to support forward protrusion. (**B**) Integrin engagement results in FAK dimerization and activation, recruiting signalling mediators that eventually disassemble point contacts and disengage the extracellular ligand. (**C**) Negative ECM guidance cues can inactivate integrins to disable point contact formation, and/or increase local activation of FAK to disassemble IACs and prevent local protrusion. (**D**) Mechanisms in (**B**,**C**) both result in point contact turnover, internalising the machinery and enabling its prompt recycling to new sites of adhesion. Previously engaged integrins from (**B)** may exhibit a conformational memory, imparted by FAK, that retains their activated state prior to presentation at the membrane.

**Table 1 biology-12-00954-t001:** Table of specific integrin heterodimers referred to in this review.

Integrin Receptor	Cell Model	Function	Reference
α1β1 α5β1	Adult rat DRG neurons	Viral expression of α-subunit promotes neurite outgrowth on laminin or fibronectin	[79]
α7β1	Adult mouse DRG neurons	α7-blocking antibody inhibits neurite outgrowth on laminin	[80]
α9β1	PC12s, adult rat DRG neurons	Viral expression of α-subunit promotes neurite outgrowth on tenascin-C	[20,83]
α5β1 αVβ3	COS7 cells	Inactivated by amino-Nogo	[84]
αVβ3	E16 mouse hippocampals	Mediates neurite outgrowth on chondroitin sulphate-D polysaccharides	[81]
αIIβ3	CHO cells	Activated by co-expression of kindlin-2 and talin	[35]

## Data Availability

No new data were created or analysed in this study. Data sharing is not applicable to this article.

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
