# Peer review of "Clutching at Guidance Cues: The Integrin–FAK Axis Steers Axon Outgrowth"

_biology, 2023, doi:10.3390/biology12070954_

Round 1

Reviewer 1 Report

In the manuscript titled “Clutching at guidance cues: the integrin-FAK axis steers axon outgrowth” Davis-Lunn and colleagues discuss the role of integrin and focal adhesion kinase signaling during axonal growth. This is an interesting review that summarizes many years of study about this specific topic. I believe the review is ready for publication and I think it will improve after addressing the following minor concerns:

1) It is well known that integrins and FAK are key factors for neuronal development. Since this review focusses on axon outgrowth, it would be interesting that the authors highlight how this review distinguishes from previous ones, or even to briefly compare the role of integrins and FAK in dendrites.

2) It would be interesting if authors include the role of other CSPGs during neurite outgrowth (besides Aggrecan), for example, the literature points out that Neurocan and Phosphacan can induce this effect as well in retinal ganglion cells in culture (Inatani, et al. 2001).

3) This reviewer values that the authors include information about axonal regeneration. However, several papers and reviews were left out from the discussion, for example: Nieuwenhuis et al, 2018; Eva et al, 2014; Sekine et al, 2022.

4) The authors refer in some cases to specific integrins and in other cases to integrins as a whole. It would be helpful if the authors include a table summarizing the integrin types and function and the cell types.

5) In line 316 the authors referred to CNS but in the following line they give as an example DRGs which are actually PNS.

Reviewer 2 Report

Davis-Lunn

Review on integrin-FAK signaling in axon outgrowth

This review covers a simplified review of focal adhesion kinase domain structure, phosphorylation, and some of the many proteins that can bind FAK. It is written in a simplified manner and the Figures/graphics are really of limited value as there is not much visual detailed information provided. 

Missing is a discussion of the FAK-related protein Pyk2 that is also highly expressed in brain and neuronal tissues.  What is conserved between the two proteins and what potential unique functions do these proteins have in the brain? At a minimum, the author should cite a recent review on Pyk2 in the brain (de Pins et al., Front Synaptic Neurosci 2021 13: 749001). There are also recent reviews on FAK structure-function relationships.

1. line 199. “This interaction is dispelled by binding of the FAK FERM domain to the b-integrin tail…” (no reference).   The authors are mixing very old published (Schaller lab) data that has not been repeated.  FAK FERM does not bind directly to b-integrin tails.

2. Line 227. “SuperFAK” again was a limited term used by the Schaller lab.  Structural studies from several groups have shown that FAK is regulated by protein conformational changes\ (see review by J. Le Coq et al. J Cell Sci 2022 Vol. 135 Issue 20).

3. Line 240.  FAK F925 mutant does not prevent movement of all cell types. Please be specific.

4. Line 310. The “PWR” insertion has no assigned function and speculation as such is misleading.

Round 2

Reviewer 2 Report

no comments